# Arctic closure as a trigger for Atlantic overturning at the Eocene-Oligocene Transition

David K. Hutchinson [1], Helen K. Coxall[1], Matt O'Regan [1], Johan Nilsson[2], Rodrigo Caballero [2] & Agatha M. de Boer[1]

The Eocene-Oligocene Transition (EOT), approximately 34 Ma ago, marks a period of major global cooling and inception of the Antarctic ice sheet. Proxies of deep circulation suggest a contemporaneous onset or strengthening of the Atlantic meridional overturning circulation (AMOC). Proxy evidence of gradual salinification of the North Atlantic and tectonically driven isolation of the Arctic suggest that closing the Arctic-Atlantic gateway could have triggered the AMOC at the EOT. We demonstrate this trigger of the AMOC using a new paleoclimate model with late Eocene boundary conditions. The control simulation reproduces Eocene observations of low Arctic salinities. Subsequent closure of the Arctic-Atlantic gateway triggers the AMOC by blocking freshwater inflow from the Arctic. Salt advection feedbacks then lead to cessation of overturning in the North Pacific. These circulation changes imply major warming of the North Atlantic Ocean, and simultaneous cooling of the North Pacific, but no interhemispheric change in temperatures.

[1] Department of Geological Sciences, Stockholm University, 10691 Stockholm, Sweden. [2] Department of Meteorology, Stockholm University, 10691 Stockholm, Sweden. Correspondence and requests for materials should be addressed to D.K.H. (email: david.hutchinson@geo.su.se)

The Eocene–Oligocene transition (EOT), ~34 Ma ago, marked a major shift in global climate towards colder and drier conditions and the formation of the first Antarctic ice sheets[1–3]. A gradual decrease in $CO_2$ is thought to be the primary driver of the transition[4–6], causing long-term cooling and increasing seasonality through the Eocene, culminating in the glaciation of Antarctica[7–9]. Deep water circulation proxies suggest that the EOT, including the preceding 1 Myr, also marked either the onset or strengthening of an Atlantic meridional overturning circulation (AMOC)[10–12]. There are two main hypotheses for what triggered the AMOC: deepening of the Southern Ocean gateways led to wind-driven upwelling of deep water in the Southern Ocean, forcing the AMOC[13,14]; and deepening of the Greenland Scotland Ridge (GSR) led to bi-directional flow over the ridge that allowed deep water to form in the North Atlantic[15]. These hypotheses have yet to be tested in a climate model that is designed with late Eocene palaeogeography and capable of capturing the ocean circulation at the EOT.

Deep sea sedimentary drift deposits indicate the presence of deep western boundary currents in the North Atlantic from the middle Eocene, suggesting the presence of deep water formation[16,17]. This is partly supported by stable isotope analyses of $\delta^{18}O$ and $\delta^{13}C$ that indicate periodic Atlantic sinking from the middle Eocene[18]. However, several Atlantic-wide compilations of stable isotopes suggest that a four-layer structure of water masses in the Atlantic, similar to the modern day, developed in the early Oligocene[10,11,19]. These analyses suggest that in the late Eocene, northern component water was confined to shallower depths and northern latitudes, whereas around the EOT, northern component water deepened and expanded across the equator, which is also supported by Neodymium isotope tracers[20]. Recent proxy data from the Labrador Sea shows that North Atlantic bottom waters were increasing in salinity and density in the three millions years prior to the EOT[12]. This salinification trend, preceding the EOT, suggests that local forcing may have enabled North Atlantic sinking independently of the Antarctic glaciation. In addition, sea surface temperature (SST) proxies indicate no significant cooling in the North Atlantic around the EOT, in contrast to major cooling in the South Atlantic[9]. This asymmetry in temperature evolution between the hemispheres is consistent with the onset or strengthening of an AMOC at the EOT.

Neodymium isotope tracers indicate that the Southern Ocean was a major source of deep water formation during the Eocene and Oligocene[21]. In contrast to today, deep water appears to have formed in the North Pacific throughout the Eocene, in combination with Southern sinking[22,23]. A recent study of neodymium isotopes throughout the Pacific suggests that North Pacific deep water formation gradually shut down from 36 Ma onwards[24]. Mechanistically, there is a competition between Pacific and Atlantic sinking modes due to salt-advection feedbacks[25–28]. Once a sinking mode is established, sinking is reinforced by the meridional overturning circulation drawing in higher salinity waters from the subtropics to the high latitudes. Salt advection between the basins then causes a lower surface salinity to develop in the high latitudes of the non-sinking basin, making it difficult to sustain sinking in both basins simultaneously[27,29]. Here we show that a shift in sinking from the Pacific to the Atlantic can be triggered by the closure of the Arctic–Atlantic gateway at the EOT.

## Results

### Closure of the Arctic–Atlantic Gateway.
The Arctic Ocean was very fresh during the Eocene, with an estimated surface salinity of around 20–25 psu in the early and middle Eocene[30,31]. This background state was punctuated by short-lived episodes of extreme freshening (<5 psu), where the freshwater dwelling *Azolla* algae bloomed across the Arctic Ocean[32]. The middle Eocene *Azolla* blooms are contemporaneous with findings of *Azolla* in the Nordic seas, implying at least periodic shallow Arctic–Atlantic exchange, with proxy evidence for relatively low salinities (brackish; 22–30 psu) in the North Sea and Greenland Basin Sea[33,34]. Assemblages of silicoflagellates and ebridians from the middle Eocene show some similarities between North Atlantic and Arctic species, indicating an Arctic–Atlantic connection during the middle Eocene[30]. Low salinity conditions were driven by an enhanced hydrological cycle in the Eocene climate combined with topographically induced river runoff[35], and the more enclosed geometry of the Arctic basin with only narrow, shallow and ephemeral seaways in the proto-Fram Strait and Barents Sea[36,37]. Geological reconstructions of the Barents Sea indicate that this region was the only viable connection between the Arctic and Atlantic in the late Eocene, and that this connection likely closed around the EOT (Fig. 1). We therefore investigate the effect of this Arctic isolation on the EOT ocean circulation.

The environmental conditions in the Arctic Ocean from the late Eocene through the Oligocene are poorly known. This time interval coincides with a prolonged hiatus (44–18 Ma) in the only drilled Cenozoic sequence from the central Arctic[38]. However, seismic stratigraphic interpretations of stacked progradational facies on the Eastern Siberian Sea slope indicate that relative sea level in the Arctic became decoupled from global eustatic variations at the EOT[39]. This isolation remained until the opening of the Fram Strait in the early Miocene[40]. The main tectonic driver for progressive isolation in the Paleogene was the northward motion of the Greenland microplate. The associated compressional tectonics, cumulatively referred to as the Eurekan deformation[41], formed a continental land bridge extending from Arctic Canada to Svalbard. By the end of the Paleogene, the Barents Sea was the only viable seaway for exchange between the Arctic and the Norwegian–Greenland Sea.

Today the average depth of the Barents Sea is 230 m, but much of its physiography was inherited from glacial erosion in the Quaternary[42], which has been overlooked in early Cenozoic reconstructions. The pre-Quaternary Barents shelf has consistently been portrayed as a subaerial platform, uplifted in the Eocene and Neogene[43]. The most recent global palaeogeographic reconstruction for the late Eocene (38 Ma) has relatively deep (200–500 m) seaways cutting across the Barents shelf providing conduits for water exchange[36]. These seaways, however, may have actually formed at a more recent age, since they occur where the largest glacial troughs exist on the shelf today (Fig. 1). Furthermore, the existence of these seaways is at odds with geological reconstructions of the Barents Sea in the late Eocene and Oligocene that portray the shelf as a subaerial denudation surface[44–46], from which 1500–2500 m of sediment was eroded in the Paleogene and Neogene[45,46] (Supplementary Fig. 1). We apply a land bridge to the Barents Sea as a simplified representation of the gradual closing of this seaway, in line with evidence of Arctic Ocean isolation at the EOT[39] (see Methods).

Several recent modelling studies of the late Eocene (38 Ma) show that the combination of a fresh Arctic Ocean and its narrow and shallow connection to the Atlantic ocean circulation prevented North Atlantic sinking[35,47,48]. In these simulations, sinking is found either in the Southern Ocean or the North Pacific or both, but never in the North Atlantic. Furthermore, a modelling study with neodymium tracers enabled found the best match to Eocene observations with strong sinking in both the North Pacific and South Pacific[23]. Modelling studies have found that closing the Arctic–Atlantic gateway can trigger North Atlantic sinking under early Eocene boundary conditions[49,50], but this mechanism has yet to be tested under EOT

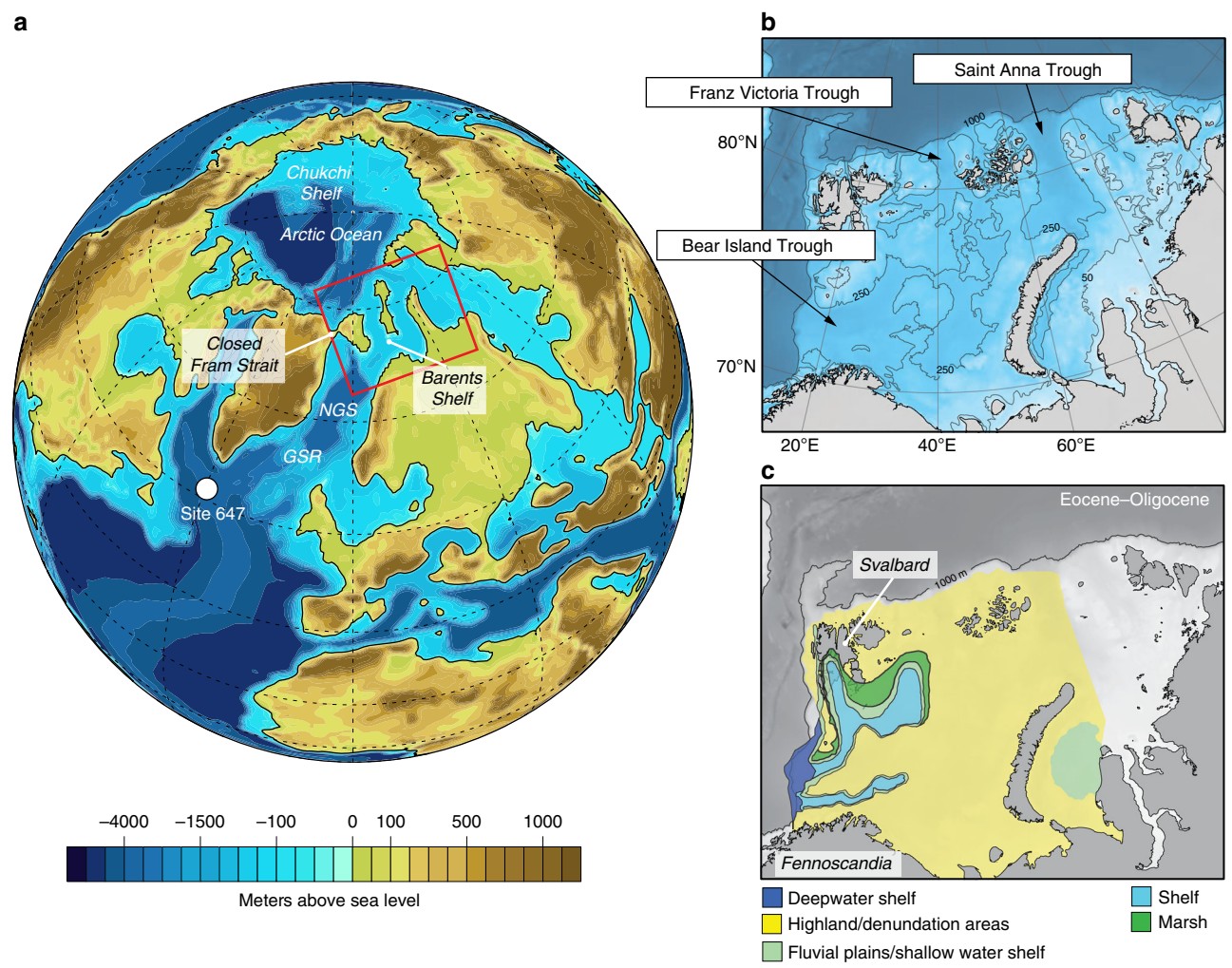

**Fig. 1** Contrasting evidence for the existence of seaways across the Barents Shelf near the Eocene–Oligocene transition (EOT). **a** Palaeogeography for the late Eocene as portrayed by Baatsen et al.[36]. **b** Modern bathymetry of the Barents shelf. The seaways present across the Barents shelf in the palaeogeography of Baatsen et al.[36] coincide with the deepened glacial troughs that were formed during Quaternary glaciations. **c** Re-drawn depositional environments on the Barents shelf in the late Eocene, published in Smelror et al.[44], p. 123. The gradual closure of seaways between Svalbard and Greenland and across the Barents Sea during the Eocene is consistent with far-field geologic evidence[30,39] and recent tectonic reconstructions of the Northwestern and Southwestern margins of the Barents Sea (between Svalbard and mainland Norway) by Lasabuda et al.[45,46]. See Supplementary Fig. 1 for further details

palaeogeography. Given the proxy evidence of the AMOC start-up at the EOT, it is important to determine how the Arctic–Atlantic gateway impacts the ocean circulation under late Eocene palaeogeography.

**Ocean and climate response to Arctic closure**. We investigate the effects of closing the Arctic–Atlantic gateway using a coupled climate model, described in the Methods. Two simulations are performed: a pre-EOT control simulation with an open Arctic–Atlantic gateway (~200 m depth; Supplementary Fig. 2) and a post-EOT simulation where the gateway is closed. In the control simulation, there is sinking in the North Pacific, but not in the North Atlantic. There is a dramatic increase in surface salinity in the North Atlantic in response to the closing of the Arctic gateway, increasing from around 25–30 psu when open to around 34–35 psu when closed (Fig. 2). The latter level of salinity is similar to that of the deep ocean and the modern-day North Atlantic. This enables the establishment of an AMOC instead of the Pacific Meridional Overturning Circulation found in the open Arctic experiment (Fig. 3). We note that sinking occurs in both the Nordic Seas and the Labrador Sea, as indicated by deep mixed

layers in these areas (Supplementary Fig. 3). This is in contrast to a previous early Eocene modelling study which found that Atlantic sinking occurred only in the Nordic Seas[50]. We hypothesise that this difference may be due to the widening of the Atlantic basin from early to late Eocene. A wider Atlantic basin increases the area of wind forcing on the North Atlantic gyres, which may alter the heat and salt transport of the western boundary currents towards the deep water formation regions.

In the closed Arctic state, the North Pacific has lower surface salinities than in the control simulation, primarily due to the salt-advection feedback[25]. This feedback introduces a competition between the two northern deep water sources, implying that when the sinking and salinity in one basin increase, the sinking and salinity in the other basin decrease[26,27]. In the control simulation, salt is transported from the Atlantic to the Pacific through the Panama Seaway, whereas in the Arctic closed experiment, this gateway transport largely disappears (Fig. 4). The competition between North Atlantic and North Pacific sinking is in line with proxy indications of a link between the onset or strengthening of Atlantic sinking[10,11] and the cessation of North Pacific sinking[24], which likely prevailed during the Eocene[23]. Once established,

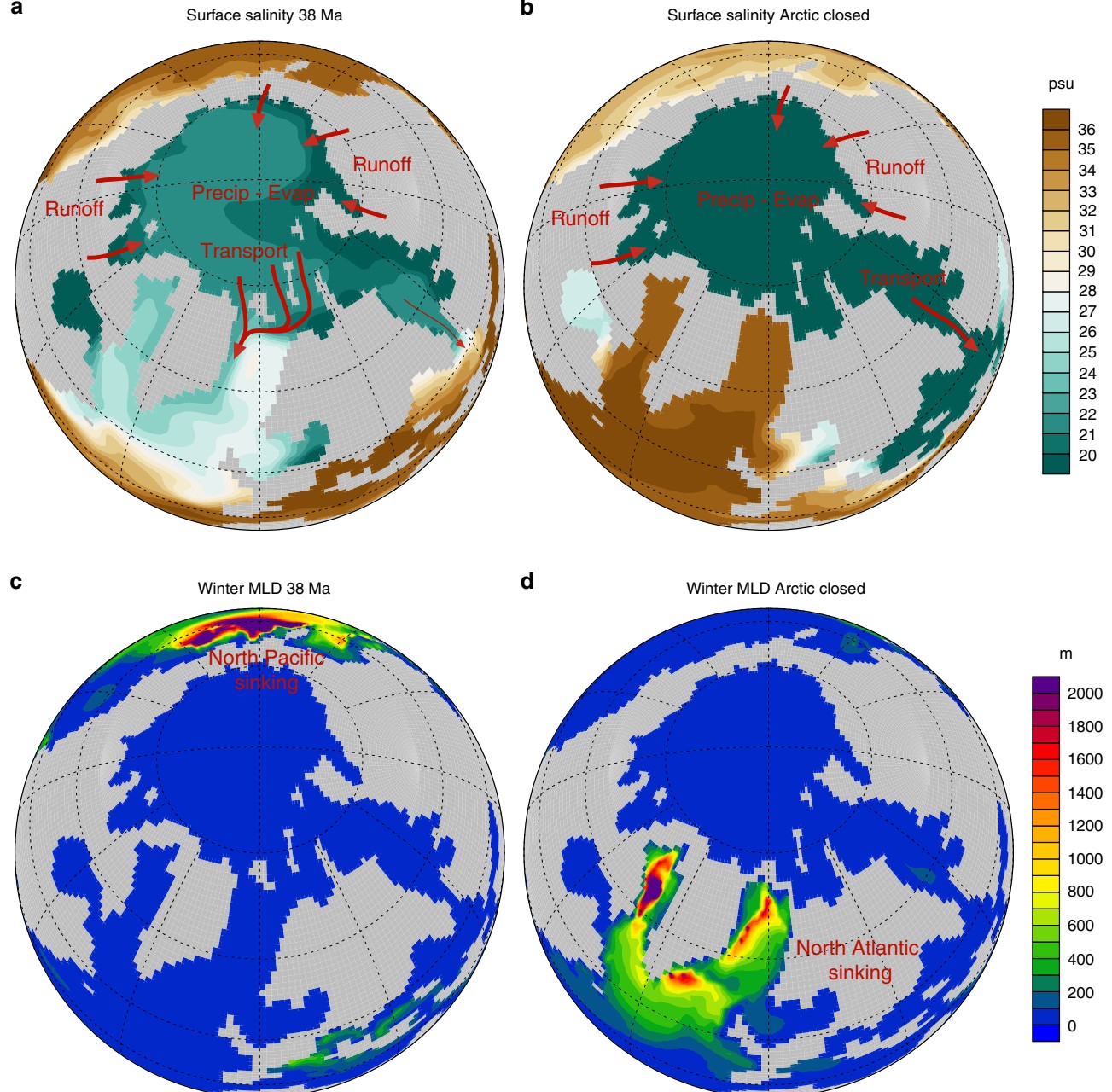

**Fig. 2** Surface salinity before and after closure of the Arctic gateway. **a** Surface salinity in the control run at 38 Ma, **b** surface salinity with the Arctic–Atlantic gateway closed. **c** Winter mixed layer depth (MLD) in the 38 Ma control run, and **d** MLD after closing the Arctic gateway. The Arctic Ocean is very fresh, due to precipitation–evaporation and continental river runoff. Closure of the Arctic–Atlantic gateway leads to a dramatic increase in North Atlantic salinity. This in turn triggers North Atlantic sinking, and the cessation of North Pacific sinking

atmospheric feedbacks enhance the salinity asymmetry due to a decrease in precipitation minus evaporation in the North Atlantic, while in the North Pacific the pattern is reversed (Fig. 5). This is due to the strong dependence of evaporation on SST. This transition demonstrates that the outflow of fresh surface water from the Arctic Ocean during the Eocene may have had a limiting effect on the AMOC. Although our experiments represent two distinct open/closed gateway experiments, in reality the closure of the gateway is likely to have been a gradual transition over several million years. Thus, a gradual closure of this gateway can explain proxy evidence that the Labrador Sea bottom salinity increased by around 5 psu over a 3–4 million year interval leading up to the EOT[12].

**Existing hypotheses for AMOC start-up**. The deepening of the GSR at the EOT has been proposed as a trigger for the AMOC, by changing the flow across the ridge from a shallow unidirectional flow to a deeper bi-directional flow[15]. According to this hypothesis, the deeper exchange allows salty subtropical water to penetrate further north and thus enable North Atlantic sinking. A recent modelling study using Miocene boundary conditions found that when the GSR was deepened to more than 50 m, an AMOC was triggered[51]. We test whether there is a similar effect under late Eocene geography, by comparing our reference topography with a deep GSR (~500 m) to a shallow GSR simulation (25 m). With an open Arctic gateway, we found that the North Atlantic remains too fresh to support sinking and sinking persists

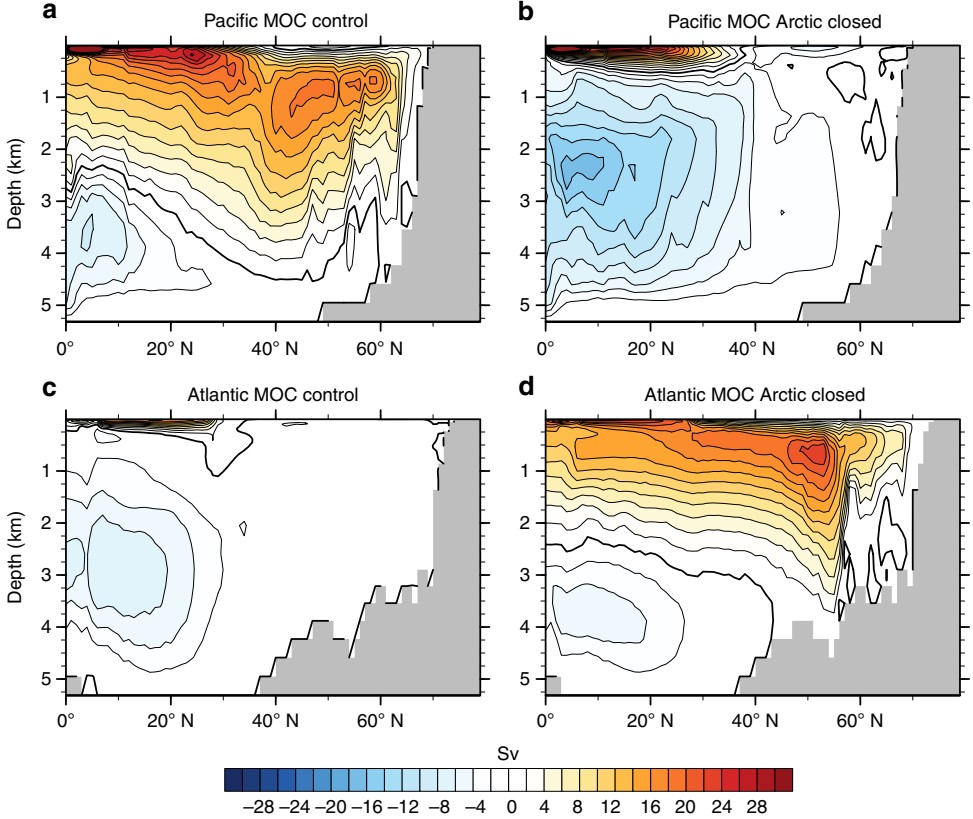

**Fig. 3** Meridional overturning circulation (MOC) in the North Pacific versus North Atlantic basins. **a** Pacific MOC in the control run, **b** Pacific MOC with the Arctic–Atlantic gateway closed, **c** Atlantic MOC in the control run and **d** Atlantic MOC with the Arctic–Atlantic gateway closed. In the control simulation, there is strong overturning in the North Pacific, while the North Atlantic remains stratified. Once the Arctic–Atlantic gateway is closed, North Atlantic overturning is triggered, while North Pacific overturning shuts down. Southern Ocean overturning is present in both simulations

in the North Pacific (Supplementary Fig. 3). We also test the shallow GSR (25 m) in the Arctic closed configuration. With the shallower GSR, sinking continues to occur in the Labrador Sea, but shifts southwards from the Nordic Seas to just south of the GSR. However, North Atlantic salinity remains high (34–35 psu), since the freshwater source from the Arctic is cut off. Thus, we found that the depth of the GSR, whether 25 or 500 m, does not change the preferred basin of sinking, despite shifting the local sinking region.

A long-hypothesised trigger of the AMOC at the EOT is the opening of Southern Ocean gateways[13]. This theory posits that when meridional land barriers are present in the Southern Ocean, midlatitude westerly winds drive a northward Ekman transport that returns southward in upper ocean western boundary currents. The opening and deepening of gateways cut off the upper ocean southward current, forcing a return flow to occur below the deepest remaining barriers in the Southern Ocean, which enables an AMOC sustained by wind-forced upwelling of deep water[52]. Modelling studies have shown that opening of the Southern Ocean gateways can trigger a switch from Southern Ocean-only to North Atlantic sinking[53,54]. However, this effect was demonstrated using modern geography with selected gateway perturbations, rather than a full palaeogeographic reconstruction.

Here, we test the effect of opening the Southern Ocean gateway upon the deep circulation at the EOT. To achieve this, we widened the Tasman gateway and the Drake Passage to 1300 and 1100 km, respectively, and deepened both gateways to 3000 m depth (grid cells already deeper than 3000 m were left unchanged). This enables an Antarctic Circumpolar Current (ACC) flow of around 150 Sv, comparable to the modern ACC.

These gateways are wider and deeper than expected at the EOT, and are designed to provide a large signal to test this hypothesis, rather than a realistic palaeogeography. Although this perturbation enhances Antarctic bottom water formation, it does not affect the preferred basin of sinking in the Northern Hemisphere. Deep water continues to form in the North Pacific, while the North Atlantic surface waters remain dominated by freshwater outflow from the Arctic Ocean (Supplementary Fig. 3). Thus, we do not find evidence that Southern Ocean gateway opening triggered the AMOC at the EOT. Furthermore, due to the long timescale and uncertainty associated with opening the Southern Ocean gateways[55], a step change in circulation is less geologically plausible than a gradual transition over tens of millions of years. By contrast, the closure of a narrow and shallow gateway such as the Arctic–Atlantic can readily be triggered within a few million years due to local uplift and changes in sea level[39,45,46].

Although the deepening of Southern Ocean gateways did not trigger an onset of Atlantic sinking in the present simulations, these gradual tectonic events may have influenced the features of a previously established AMOC mode. Specifically, deepening of the gateways is expected to reinforce the AMOC through wind-forced Southern Ocean upwelling[52]. Thus, it is possible that the AMOC had strengthened by the time the Arctic–Atlantic connection reopened, estimated to have occurred during the early Miocene[40], allowing North Atlantic sinking to persist despite an increased Arctic Ocean freshwater influx. This underlines that the global bathymetry and climate state may be critical for the response of the AMOC to local gateway changes in the northern North Atlantic.

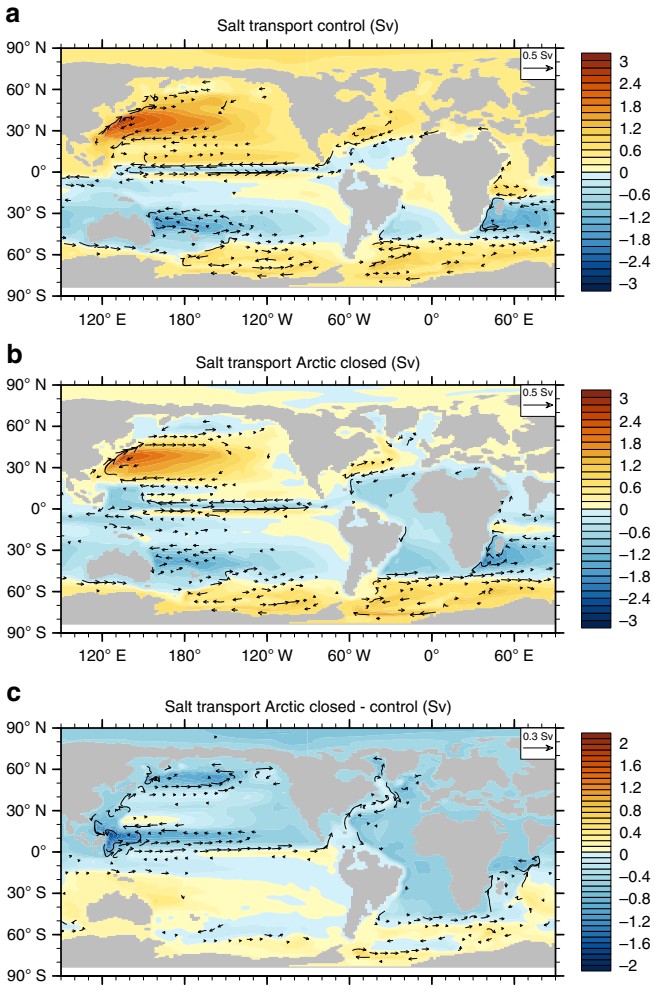

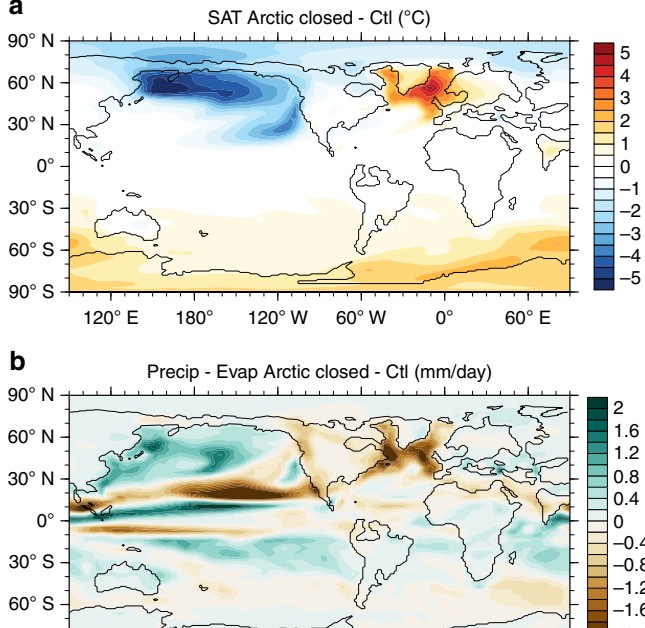

**Fig. 5** Temperature and freshwater flux response to closing the Arctic gateway. **a** Surface air temperature (SAT) and **b** precipitation minus evaporation differences showing the Arctic closed—control experiment in each case. There is substantial cooling (>5 °C) of the North Pacific in response to the shutdown of Pacific sinking, and a corresponding warming of the North Atlantic. No major interhemispheric temperature change is found. Precipitation minus evaporation increases in the North Pacific and decreases in the North Atlantic

**Fig. 4** Depth-integrated salt transport streamfunction. **a** The control experiment, **b** the Arctic Closed experiment and **c** the difference between Arctic Closed—Control. Contour intervals are 0.2 Sv, while transport vectors are drawn above a threshold magnitude of 0.05 Sv in **a**, **b** and 0.03 Sv in **c**. The presence of overturning in the North Pacific creates greater salt transport into the high latitude sinking regions, along with westward salt transport through the Panama gateway. Once the AMOC is triggered, salt is drawn into the high latitudes of the North Atlantic, causing a near shutdown of salt transport through the Panama gateway

Previous modelling studies have found that Arctic closure can trigger an AMOC under early Eocene palaeogeography[49,50]; however, this is substantially different from the late Eocene. The North Atlantic basin widens over the 20 million years from the early to late Eocene, which affects the pathways of heat and salt transport to the possible sinking locations. In particular, a wider Atlantic basin creates a larger basin of wind forcing for the North Atlantic gyres and more prominent western boundary currents. It was therefore not obvious that the Arctic closure would trigger an AMOC using late Eocene palaeogeography, nor that the AMOC would be shut down when the Arctic gateways are open. Furthermore, by testing our results in combination with the Southern Ocean opening and GSR hypotheses, we have demonstrated that the Arctic Ocean gateway closure is more important than other hypothesised triggers of the AMOC at the EOT.

## Discussion

Here we have argued that the onset of modern-like deep water formation in the North Atlantic was triggered by the gradual closing of the Arctic from the Atlantic prior to the EOT. The question remains whether this geographically induced AMOC start-up is causally related to the Antarctic glaciation at the EOT. We found that Arctic closure could not have induced an Antarctic ice sheet through heat transport changes alone because the Antarctic continent actually warms by an average of 1.4 °C after closure. Southern Ocean temperature proxy records do in fact detect warming on the order of 1–2 °C in the latest Eocene[56], consistent with recent suggestions that the AMOC strengthened ~1 to 0.5 Myr prior to Antarctic glaciation[12]. If Arctic closure played a role in Antarctic ice sheet formation, it must have occurred through indirect effects, such as circulation changes lowering atmospheric $CO_2$, or altering precipitation over Antarctica. We found that a halving of $CO_2$ from 800 to 400 ppm induces a cooling of Antarctica of around 8 °C (Supplementary Fig. 4). This is likely an overestimate of the $CO_2$ drop across the EOT; however, it illustrates that the climate impacts of $CO_2$ forcing on Antarctica are likely to dominate over ocean heat transport effects induced by the change in circulation.

Atmospheric $CO_2$ can decrease due to many factors, such as a reduction in $CO_2$ emissions, or changes to geological sinks or carbon reservoirs. We consider here two potential mechanisms of atmospheric $CO_2$ reduction, which could be related to the modelled circulation changes, namely, increased ocean carbon uptake and enhanced weathering. We found that the global mean ventilation age of the ocean increases from 479 to 570 years. This longer ventilation time may imply a greater capacity to store carbon in the deep ocean. Tentative estimates of circulation-induced changes to deep ocean carbon storage capacity suggest that such changes imply a relatively small change in atmospheric $CO_2$ (~10–30 ppm)[57] compared with the overall decline in

atmospheric $CO_2$ over the Eocene–Oligocene of several hundred ppm[5]. The other proposed mechanism to reduce atmospheric $CO_2$ is enhanced silicate weathering. Elsworth et al.[54] argued that a Southern Ocean gateway-triggered AMOC start-up led to an increase in precipitation and temperature in the Northern Hemisphere, which caused a drawdown of $CO_2$ through enhanced silicate weathering. We do not find evidence for this mechanism, because the increased precipitation in the Atlantic is balanced by a decrease in precipitation in the Pacific (Fig. 5). We also do not find evidence for a northward shift in the ITCZ, as suggested by a change in dust provenance in the eastern equatorial Pacific from Asian sources to South/Central American sources[58] (Supplementary Fig. 5). This may be because we do not have an ice sheet in our model and these changes are interpreted to be a response to Antarctic glaciation and asymmetric polar cooling[59]. However, we found a narrowing of the northern branch of the Pacific ITCZ, associated with a strengthening of the subtropical anticyclone in the North Pacific. The latter is likely related to SST cooling in the North Pacific and resulting increase in land–ocean temperature contrast[60].

In order to properly understand the links between overturning changes and changes in $CO_2$, further research is needed using climate models that include a carbon cycle and weathering processes[61]. Nevertheless, our results provide a strong motivation to better understand the physical climate and ocean circulation changes at the EOT in the context of a well-resolved palaeogeography. The close timing between this overturning shift and the EOT glaciation suggests that the two events could be causally related.

## Methods

**Climate model simulations**. The simulations were performed using the GFDL CM2.1 model adapted to late Eocene (38 Ma) boundary conditions. The control simulation, using atmospheric $CO_2$ of 800 ppm, is described in detail in Hutchinson et al.[35]. The model uses the palaeogeography reconstruction of Baatsen et al.[36], with an ocean resolution of $1° \times 1.5° \times 50$ levels and an atmosphere resolution of $3° \times 3.75° \times 24$ levels. This allows better representation of ocean gateways than most existing Eocene–Oligocene climate models, which typically use lower ocean resolution, or employ geographic boundary conditions not designed for this time interval. The Arctic closed experiment uses the same configuration except that the Arctic–Atlantic gateway is closed by a land bridge, in line with geological evidence in the lead up to the EOT. Furthermore, we have closed the narrow passageway in our control simulation between Greenland and Svalbard to ensure Arctic isolation. We note that this passageway between Greenland and Svalbard was an artefact of re-gridding the Baatsen et al.[36] dataset onto our model grid, and was closed in the original dataset (Fig. 1a). Sensitivity tests indicate that this artefact has minimal impact on the North Atlantic salinity compared with the wider and deeper open Barents Sea connection, and our control salinity climatology is similar to that found in Baatsen et al.[48]. The Arctic Ocean retains a shallow outflow through the Turgai Strait into the Tethys Ocean, which ensures that Arctic freshwater accumulation does not cause numerical instabilities. This outflow has negligible impact on Atlantic water mass properties.

The control simulation has a similar horizontal ocean circulation to the model configuration of Baatsen et al.[47,48], who explore the ocean circulation in an ocean-only and a fully coupled climate model forced using the same late Eocene palaeogeography as this study. Surface air temperature and precipitation patterns are also similar to Baatsen et al.[48]. A key difference between our model and theirs is that we find bipolar Pacific sinking, whereas they find unipolar Pacific sinking. Their ocean-only model simulations exhibit either North Pacific sinking or South Pacific sinking, depending on the boundary and initial conditions[47], while their coupled model exhibits South Pacific sinking under the same palaeogeography[48]. These differences may be due to differences in model resolution and re-gridding effects, or differences in internal dynamics of each model. Their model uses a resolution of ~1° in the ocean (with a different discretization) and ~2° in the atmosphere, giving rise to differences in the resulting topography, runoff and freshwater forcing in the sinking regions. Nevertheless, our simulations broadly support their findings that both on and off states of North Pacific sinking can be found under this palaeogeography, and the North Atlantic is too fresh to sustain sinking with an open Arctic gateway. Given that our model exhibits distinct states of North Pacific and North Atlantic sinking, governed by salt-advection feedbacks, we cannot exclude the possibility of bistability. However, this would require significant further testing and re-equilibration of the model and is beyond the scope of this study.

**Spin-up and perturbation experiments**. The closed Arctic experiment was spun up for 6500 years from idealised initial conditions using an iterative coupling procedure identical to the control experiment, with the last 3200 years being run in fully coupled mode. Time series of SST and temperatures at 2000 and 4000 m depth are shown in Supplementary Fig. 6, along with MOC indices, and North Atlantic surface temperature and salinity. The deep ocean is still cooling after 6500 years, but the overturning circulation is steady in both hemispheres, and the surface temperature and salinity in the deep water formation regions are stable.

The wide Southern Ocean gateways and shallow GSR experiments were branched from the control run at year 5500, and run for a further 1000 and 1500 years, respectively. The shallow GSR experiment was extended due to a transient shutdown of global overturning during re-equilibration; thereafter, sinking was re-established in the North Pacific and the Southern Ocean (Supplementary Figure 3). The shallow GSR–Arctic closed experiment was branched from year 5500 of the Arctic closed experiment and run for a further 500 years. No substantial changes to the overturning circulation were found. Changes to the bathymetry in the perturbation experiments are shown in Supplementary Fig. 2.

## Data availability

The model data presented in this study is available for download from the Bolin Centre database, at https://bolin.su.se/data/hutchinson-2019.

## Code availability

Model code for CM2.1, using the ocean model version MOM 5.1.0, is available for download at https://mom-ocean.github.io/.

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

## Acknowledgements

This study was supported by the Swedish Research Council Project 2016-03912, FORMAS Project 2018-01621 and the Bolin Centre for Climate Research, Research Areas 1 and 6. Numerical simulations were performed using resources provided by the Swedish National Infrastructure for Computing (SNIC) at NSC, Linköping. Open access funding provided by Stockholm University.

## Author contributions

D.K.H. and A.M.d.B. led the study and designed the experiments. D.K.H. conducted the experiments and wrote the first draft. H.K.C. provided proxy evidence of deep water circulation, M.O'R. undertook the palaeogeography analysis, J.N. analysed the inter-basin salt transport and R.C. analysed the atmospheric circulation. All authors contributed to the analysis and editing of the manuscript.

## Additional information

**Competing interests:** The authors declare no competing interests.

