## [Peer Review File · Nature Communications]

Reviewers' comments:

Reviewer #1 (Remarks to the Author):

The manuscript by Hutchinson et al., addresses the behavior and drivers of the Atlantic Meridional Overturning Circulation (AMOC) a theme of great scientific and future societal relevance aligned with Nature Communications profile. The authors present the first study utilizing a paleogeography for the late Eocene, testing the hypothesis that a closure of the Arctic-Atlantic gateway at this time hindered freshwater inflow from the Arctic to the deep-water formation areas in the Nordic Seas and Labrador Sea, thus triggering the initiation of North Atlantic Deep Water formation and the cessation of deep water formation in the North Pacific.

Various circulation proxies suggest that the AMOC initiated either during the early to middle Eocene (at ~49 Ma and invigorated at the EOT) or initiated about 15 million years later during the EOT (~34 Ma). While a number of similar modeling studies have addressed causes for earlier time slices (Roberts et al., 2009, Vahlenkamp et al., 2018), this study is to my knowledge the first that addresses gateway changes at the EOT with a fully coupled Earth System Model. Some of the findings (e.g. dependence of deep-water formation in the Nordic Seas during the Eocene on limiting the inflow of fresh Arctic surface waters) have been previously reported by studies that used an early Eocene continental setup. Others (e.g. the switch from Pacific to Atlantic overturning) are new and exciting and are here interpreted in the context of the large-scale environmental changes at the EOT. The discussion of results in the context of the changes at the EOT is interesting.

I have no concerns about methods and believe that the results largely support conclusions. The manuscript is well designed, contains a number of interesting observations and I found it enjoyable to read.

General comments

It is critical that the intermediate and deep oceans are fully equilibrated for evaluation of ocean currents. While the run times seem generally long enough, no evidence is presented here. The readers should get a chance to evaluate the level of equilibrium in form of timeseries of deep and intermediate ocean temperatures as an Extended Data Figure or at least a statement on the temperature drift (something similar can be found in e.g. Roberts et al., 2009).

One interesting difference between this study and those for the early Eocene is that strong overturning occurs in the Labrador Sea during the late Eocene, whereas in early Eocene simulations the Nordic Seas are the major source of NCW/NADW. A discussion of this phenomenon and the possibility that some form of NCW initiated in the Nordic Seas during the early-middle Eocene and reached near modern strength with the onset of additional Labrador Sea overturning at the EOT could be an asset of this manuscript.

If the format of the article allows for it, I think the manuscript would benefit from shifting some figures from the supplements to the main body. Robust evidence for the occurrence of these gateway changes at this time is a crucial factor in this manuscript, so it would be nice to move Extended Data Figure 3 to the main manuscript. I also would consider moving Extended Data Figure 1 to the main manuscript

Specific comments

Page 1 line 13, Page 2 line 5 and other occasions: Here the authors refer to the “onset of the AMOC” at the EOT. I think this needs some further elaboration/definition to what exactly is meant by the onset of the AMOC as the term AMOC and its different components are quite a complex construct by itself. E.g. as the authors mention on page 2 lines 17-21 there is evidence that suggests strong Deep Western Boundary Currents from ~49 Ma onwards. This implies that at least some components of the AMOC (if the AMOC is defined as the Atlantic component of the “global conveyor belt”) were active in some form long before the EOT.

Page 2 lines 23-25 – Page 3 lines 1-3: References? Or is this already an outcome of this study?

Page 3 lines 10-12: Figure 3 shows that the high Southern latitudes warm quite a bit due to the onset of the AMOC. Is there any proxy evidence from these areas that shows muted cooling at the EOT to additionally support this statement?

Page 5 line 16: Odd section break starting of the sentence with “This transition...”

Page 6 line 7-9: Actually, there is one fundamental change between the two GSR scenarios with a closed Arctic (Figure 2b and 2d): the overturning in the Nordic Seas is completely absent when the GSR is shallow, while North Atlantic and Labrador Sea overturning are not affected. This should be mentioned.

Extended Data Figure 1: Should specify in the Figure caption whether these are sea surface maps. Particularly as the surface currents are flowing southward in the western South Atlantic. This comes back to the question of defining what the onset of the AMOC actually means. Southward flow of surface waters in the western South Atlantic (Extended Data Figure 1b) is not a feature common feature of the modern AMOC.

Reviewer #2 (Remarks to the Author):

Review of Hutchinson et al, "Arctic closure as a trigger for Atlantic overturning at the Eocene-Oligocene Transition"

In this paper, Hutchinson et al present model simulations of the late Eocene with various gateway configurations. In particular, they explore the role of the Atlantic-Arctic connection, through "open" and "closed" simulations. They find that with an open gateway (taken to represent prior to the EOT) the North Atlantic is relatively fresh, which inhibits deep water formation. With a closed gateway (post EOT), the salinity of the North Atlantic increases, deep water commences, and the AMOC initiates. Conversely they find that varying the depth of the GSR, or width/depth of the southern ocean gateways, has little effect on the AMOC.

This is a very well written paper, with clear figures, and with a simple and alluring result.

General comments:

The paper as currently framed does rather rely on the geological evidence for gateway closure at this time. Although this is (somewhat cursorily) tackled in Methods, I am not convinced. It may be better to pull back on the significance of this mechanism at a specific time, and instead present it more as a sensitivity study.

More information is needed on the spinup of the simulations, and this should be illustrated in the Extended Data. For example, timeseries of global mean surface temperature, and timeseries of local surface temperature and salinity in the region of the North Atlantic should all be shown. This is essential for verifying if the results are a transitory feature or are in equilibrium. Also, timeseries in the deep ocean will allow assessment of the equilibrium state of the large scale circulation.

I think that the final section on implications for glaciation (really a section on implications for CO₂ via weathering changes) is rather weak. The weathering process is so complex that I don't think that any robust conclusion can be drawn by just examining global or even regional precipitation/temperature changes. Weathering is such a non-linear process that this section is rather meaningless, in my

opinion. It could be possible to couple the results to a weathering model, such as Geoclim (Godderis et al), but this would likely be beyond the scope of this manuscript so I suggest cutting this section entirely. However, cutting this section would leave the paper a little “thin” for Nature Communications, in my opinion.

I think that there should be more comparison with the work of Baatsen et al, which uses an identical paleogeography. How do his ocean circulation states compare with yours, and do his results also support your conclusions? In several of his published papers, there are comparable simulations that focus on ocean circulation and I am surprised that more explicit comparisons with them are not made.

After the EOT, the Arctic gateway clearly opened again at some point, but the Arctic did not freshen and the AMOC didn't shut down again. Why was that? Was it because the Bering Strait opened, and that allowed the Arctic to become more saline?

Can you hypothesise why the GSR mechanism appears to be important in the Miocene but not here? (or are the Starz et al simulations flawed in some way?)

Specific comments:

Page 2, Line 5: Page 2, line 22: Figure 3 of Ferrara et al implies that the proxies indicate a more complex history of NADW than just “turning on” at the EOT. SO would be good to moderate the language here somewhat.

Page 3, Line 11: not sure why this is a “increase in temperature asymmetry”. There is an asymmetry, yes, but not sure why it is increasing.

Page 4, line 1: “Geological reconstructions of the Barents Sea indicate that this region was the only viable connection between the Arctic and Atlantic in the late Eocene”. What about east of Greenland? What about via Turgai Strait and paratethys?

Page 6, line 22 “northward Ekman transport returns geostrophically” I am not sure what “returns” means in this context.

P7, line 16. Just because southern ocean gateway are uncertain, does not mean that they are less plausible as a mechanism.

Reviewer #3 (Remarks to the Author):

This manuscript reports a paleosimulation follow up study to Coxall et al., 2018 (NGS). The authors' aim is to investigate the cause of the inception of Atlantic overturning circulation (AMOC) at the Eocene Oligocene transition. They conclude in favour of Arctic closure as the trigger as opposed to the two more established hypotheses- deepening of the Greenland Scotland Ridge and development of wind-driven upwelling in the Southern Ocean.

We don't know that AMOC kicked in at the EOT but there are strong hints in published data sets that AMOC at least strengthened at that time. That's potentially a big deal because we do know that Antarctica grew its first large ice sheets at the EOT.

The manuscript falls short of establishing a mechanistic link between AMOC and Antarctic glaciation but that's okay. There's enough here to merit publication subject to some improvements outlined below. I think that these issues can be addressed without a big inflation in manuscript size because there are space savings to be made. For example the introduction could be tightened up by ditching the review text which goes back over elderly composite data sets/reviews. They could instead drive more rapidly to the key proxy evidence from the Atlantic for EOT AMOC start up.

We need an improved project justification/articulation of the study's major advance. What, more precisely, is the original/significant result here? It is a useful step and important contribution to have shown that Arctic isolation promotes strong densification of and deep convection in the North Atlantic but, as the authors declare, that is not an entirely new result (ref 37 38). Given that the mechanism responsible for this change is gateway closure to the Arctic, it is not obvious why we can't confidently apply the findings from those Early Eocene studies to the EOT. What are the differences, say in continental configuration/hydrological cycle between the Early Eocene and EOT or numerical models used that justify this new study and make its results unanticipated? A related issue is that we need a more prominent critical evaluation of the tectonic case for Arctic isolation around EOT time. There's no reference supporting the Barents Sea argument in the main text and too much

of that discussion is hidden away in the Methods. What are the realistic uncertainties (including chronological) on the interpretations of refs 50-52?

The authors should tighten up on their terminology and address interesting results in their experiments (SAT) that contradict the present day. What exactly is a modern-like AMOC? It's a term used liberally in the manuscript but I can't find the authors' definition of what they mean. The text that comes closest is the discussion in paragraph-2 (bottom half of p.2) but it needs work. There's a garbled statement about composite stable isotope data sets and, in my view, it is ill-advised to state as a fact that modern-like AMOC developed at the EOT based on those data. More importantly, I find the result presented in Figure 3a interesting but doesn't it fundamentally contradict the idea that the inception/invigoration of AMOC at the EOT is in anyway modern-like? Today the northern hemisphere is warmer than the southern hemisphere largely because of AMOC-driven northward cross-equatorial heat transport. Figure 3a suggests that, if the continents are configured for the EOT, AMOC-driven warming of the circum-North Atlantic is completely counteracted by cooling in the North Pacific because of shutdown of its deep convection cell.

Figure 3b (P-E) is also interesting but under-explored in discussion. The authors focus a little obsessively on their P-E field results for the continents, with the aim I think to address the specific chain of events hypothesized by Ref 42 to link AMOC and Antarctic glaciation. But there appears to be a very nice result in their P-E simulations over the ocean that goes undiscussed. I see a strong result in the (sub)tropical belt, especially over the Pacific that appears diagnostic of an intensified/shifted ITCZ that merits more attention (cf. Hyeong et al., 2016).

Finally, I'd really like to see a further panel with polar projections for Figure 3 to better evaluate the footprint of changes in SAT and P-E on the real estate that became/did not become glaciated at the EOT (Antarctica/Arctic).

Details

Sort out the line numbering, 1-25 on every page is painful.

Bold para line 4, it is ambiguous

Bold para the last sentence needs to be bigger picture- what are the implications?

Page 3, para-1 this transition is ambiguous

Page 4, line 25 and overpage- sentence is garbled

Paul A. Wilson

We thank the reviewers for their constructive comments which have helped to substantially improve the manuscript. We outline our response to each of the comments in blue text below.

Kind Regards,
David Hutchinson

Reviewer #1 (Remarks to the Author):

The manuscript by Hutchinson et al., addresses the behavior and drivers of the Atlantic Meridional Overturning Circulation (AMOC) a theme of great scientific and future societal relevance aligned with Nature Communications profile. The authors present the first study utilizing a paleogeography for the late Eocene, testing the hypothesis that a closure of the Arctic-Atlantic gateway at this time hindered freshwater inflow from the Arctic to the deep-water formation areas in the Nordic Seas and Labrador Sea, thus triggering the initiation of North Atlantic Deep Water formation and the cessation of deep water formation in the North Pacific.

Various circulation proxies suggest that the AMOC initiated either during the early to middle Eocene (at ~49 Ma and invigorated at the EOT) or initiated about 15 million years later during the EOT (~34 Ma). While a number of similar modeling studies have addressed causes for earlier time slices (Roberts et al., 2009, Vahlenkamp et al., 2018), this study is to my knowledge the first that addresses gateway changes at the EOT with a fully coupled Earth System Model. Some of the findings (e.g. dependence of deep-water formation in the Nordic Seas during the Eocene on limiting the inflow of fresh Arctic surface waters) have been previously reported by studies that used an early Eocene continental setup. Others (e.g. the switch from Pacific to Atlantic overturning) are new and exciting and are here interpreted in the context of the large-scale environmental changes at the EOT. The discussion of results in the context of the changes at the EOT is interesting.

I have no concerns about methods and believe that the results largely support conclusions. The manuscript is well designed, contains a number of interesting observations and I found it enjoyable to read.

We thank the reviewer for the positive feedback.

General comments

It is critical that the intermediate and deep oceans are fully equilibrated for evaluation of ocean currents. While the run times seem generally long enough, no evidence is presented here. The readers should get a chance to evaluate the level of equilibrium in form of timeseries of deep and intermediate ocean temperatures as an Extended Data Figure or at least a statement on the temperature drift (something similar can be found in e.g. Roberts et al., 2009).

We now include spinup time series of ocean temperature at the surface, 2000 m and 4000 m, as well as MOC indices and North Atlantic surface temperature and salinity in Extended Data Figure 1. The deep ocean is still cooling after 6500 years, but the overturning circulation is steady in both hemispheres, and the surface temperature and salinity in the deep water formation regions are stable.

One interesting difference between this study and those for the early Eocene is that strong overturning occurs in the Labrador Sea during the late Eocene, whereas in early Eocene simulations the Nordic Seas are the major source of NCW/NADW. A discussion of this phenomenon and the possibility that some form of NCW initiated in the Nordic Seas during the early-middle Eocene and reached near modern strength with the onset of additional Labrador Sea overturning at the EOT could be an asset of this manuscript.

We have added the following comment:

“We note that sinking occurs in both the Nordic Seas and the Labrador Sea, as indicated by deep mixed layers in these areas (Extended Data Figure 2). This is in contrast to a previous early Eocene modelling study which found Atlantic sinking occurred only in the Nordic Seas (Vahlenkamp et al. 2018). We hypothesise that this difference may be due to the widening of the Atlantic basin from early to late Eocene, enabling a larger region of deep water formation.”

If the format of the article allows for it, I think the manuscript would benefit from shifting some figures from the supplements to the main body. Robust evidence for the occurrence of these gateway changes at this time is a crucial factor in this manuscript, so it would be nice to move Extended Data Figure 3 to the main manuscript. I also would consider moving Extended Data Figure 1 to the main manuscript.

Thanks for this suggestion. The geological evidence for gateway closure is now included in the main manuscript, and the accompanying figure (previously Extended Data Figure 3) is now modified as Figure 1 of the main text. We have also moved Extended Data Figure 1, which is now Figure 5.

Specific comments

Page 1 line 13, Page 2 line 5 and other occasions: Here the authors refer to the “onset of the AMOC” at the EOT. I think this needs some further elaboration/definition to what exactly is meant by the onset of the AMOC as the term AMOC and its different components are quite a complex construct by itself. E.g. as the authors mention on page 2 lines 17-21 there is evidence that suggests strong Deep Western Boundary Currents from ~49 Ma onwards. This implies that at least some components of the AMOC (if the AMOC is defined as the Atlantic component of the “global conveyor belt”) were active in some form long before the EOT.

We have clarified our description of the modern-like Atlantic structure to:

“several Atlantic-wide compilations of stable isotopes suggest that a four-layer structure of water masses in the Atlantic, similar to the modern day, developed in the early Oligocene (Cramer et al. 2009; Katz et al. 2011; Borrelli et al. 2014). These analyses suggest that in the late Eocene, northern component water was confined to shallower depths and northern latitudes, whereas around the EOT, northern component water deepened and expanded across the equator.”

Page 2 lines 23-25 – Page 3 lines 1-3: References? Or is this already an outcome of this study?

We added references to idealised model studies that illustrate the inter-basin salt advection mechanism. We removed the reference to the Panama gateway here, since this is a result of our study, which we describe later.

Page 3 lines 10-12: Figure 3 shows that the high Southern latitudes warm quite a bit due to the onset of the AMOC. Is there any proxy evidence from these areas that shows muted cooling at the EOT to additionally support this statement?

The warming signal (~ 1°C) in the high southern latitudes is smaller than that induced by CO₂ cooling and glaciation, therefore, it's possible that such a warming signal might be countered by the background cooling. On the other hand, deep sea and surface ocean temperatures proxy records derived from Southern Ocean foraminiferal Mg/Ca (Bohaty et al., 2012) detect warming on the order of 1-2°C in the latest Eocene between 34.5-34 Ma. This supports the idea of southern warming associated with simultaneous strengthening of southern overturning and heat transport, because the variety of proxy data produced by

Coxall et al., (2018) imply that strengthening of NCW occurred 1 to 0.5 Myr before the onset of Antarctic glaciation. We have added a new Extended Data Figure 4 showing a polar projection of Antarctic temperature change, together with the corresponding change induced by halving CO₂. The CO₂-induced cooling combined with the effects of glaciation on Antarctica would dominate. We have also added a sentence referring to the Bohaty et al., (2012) proxy evidence.

Page 5 line 16: Odd section break starting of the sentence with “This transition...

The paragraph break has been removed.

Page 6 line 7-9: Actually, there is one fundamental change between the two GSR scenarios with a closed Arctic (Figure 2b and 2d): the overturning in the Nordic Seas is completely absent when the GSR is shallow, while North Atlantic and Labrador Sea overturning are not affected. This should be mentioned.

The shift in sinking location to just south of the GSR has now been mentioned.

Extended Data Figure 1: Should specify in the Figure caption whether these are sea surface maps. Particularly as the surface currents are flowing southward in the western South Atlantic. This comes back to the question of defining what the onset of the AMOC actually means. Southward flow of surface waters in the western South Atlantic (Extended Data Figure 1b) is not a feature common feature of the modern AMOC.

These figures represented depth-integrated salt transport, which has been clarified in the caption. We have also redrawn the figures as a streamfunction rather than the transport magnitude as before, which better illustrates the orientation and circulation patterns.

Reviewer #2 (Remarks to the Author):

Review of Hutchinson et al, “Arctic closure as a trigger for Atlantic overturning at the Eocene-Oligocene Transition”

In this paper, Hutchinson et al present model simulations of the late Eocene with various gateway configurations. In particular, they explore the role of the Atlantic-Arctic connection, through “open” and “closed” simulations. They find that with an open gateway (taken to represent prior to the EOT) the North Atlantic is relatively fresh, which inhibits deep water formation. With a closed gateway (post EOT), the salinity of the North Atlantic increases, deep water commences, and the AMOC initiates. Conversely they find that varying the depth of the GSR, or width/depth of the southern ocean gateways, has little effect on the AMOC.

This is a very well written paper, with clear figures, and with a simple and alluring result.

We thank the reviewer for the positive feedback.

General comments:

The paper as currently framed does rather rely on the geological evidence for gateway closure at this time. Although this is (somewhat cursorily) tackled in Methods, I am not convinced. It may be better to pull back on the significance of this mechanism at a specific time, and instead present it more as a sensitivity study.

The geological evidence for gateway closure is now included in the main manuscript, and the accompanying figure (previously Extended Data Figure 3) is now modified as Figure 1 of the main text. There are two important points that should be emphasized;

- 1) We summarise several lines of published geological evidence for the isolation of the Arctic near the EOT. These include sequence stratigraphic interpretations of reflection seismic data from the interior Arctic; tectonic reconstructions that illustrate collision and mountain building along the Canadian/Greenland/Barents margin during the independent northward migration of the Greenland microplate; and the most recent paleogeographic reconstructions of the Barents shelf based on extensive industry seismic and borehole data. All of these suggest that the Arctic became more isolated near the EOT and remained in this state until the early Miocene.
- 2) At the same time, we try to respectfully highlight what we see as a fundamental problem with existing Arctic paleogeographic reconstructions of the Arctic near the EOT. Specifically, these reconstructions tend to use modern bathymetry to define the most likely locations of shallow seaways in the past. In the Barents Sea, this has resulted in the EOT connections between the Arctic and the Atlantic existing in areas that were glacially eroded in the Quaternary.

While there is some uncertainty over the timing for closure of the Barents seaway, we note that the geological evidence for other popular hypotheses for AMOC startup at the EOT are not conclusive either in their timing. Furthermore, we also show that these alternate explanations cannot explain the AMOC startup when tested in our late Eocene model, so another mechanism is needed. In this manuscript we present a viable alternative, that is based on the most up-to-date paleogeographic evidence for Arctic seaways.

More information is needed on the spinup of the simulations, and this should be illustrated in the Extended Data. For example, timeseries of global mean surface temperature, and timeseries of local surface temperature and salinity in the region of the North Atlantic should all be shown. This is essential for verifying if the results are a transitory feature or are in equilibrium. Also, timeseries in the deep ocean will allow assessment of the equilibrium state of the large scale circulation.

As noted above, we have added time series of temperature evolution at the surface, 2000 m and 4000 m depth, together with MOC indices, and surface temperature and salinity in the North Atlantic in Extended Data Figure 1.

I think that the final section on implications for glaciation (really a section on implications for CO₂ via weathering changes) is rather weak. The weathering process is so complex that I don't think that any robust conclusion can be drawn by just examining global or even regional precipitation/temperature changes. Weathering is such a non-linear process that this section is rather meaningless, in my opinion. It could be possible to couple the results to a weathering model, such as Geoclim (Godderis et al), but this would likely be beyond the scope of this manuscript so I suggest cutting this section entirely. However, cutting this section would leave the paper a little "thin" for Nature Communications, in my opinion.

We have edited this section now. As argued by others, a CO₂ drop is needed together with the circulation changes to induce the glaciation. We briefly discuss ocean uptake of CO₂ and weathering as possible feedback mechanisms. We point out that an AMOC-induced weathering feedback, proposed by Elsworth et al (2017) using modern paleogeography, does not work at the EOT because Atlantic sinking occurred at the expense of Pacific overturning. Thus we maintain that our results serve as a strong motivation to accurately represent paleogeography in EOT climate simulations. We have edited this section to better acknowledge the wider climate implications, and have added Extended Data Figure 4 on Antarctic climate change induced by the overturning changes compared to CO₂-induced climate change.

I think that there should be more comparison with the work of Baatsen et al, which uses an

identical paleogeography. How do his ocean circulation states compare with yours, and do his results also support your conclusions? In several of his published papers, there are comparable simulations that focus on ocean circulation and I am surprised that more explicit comparisons with them are not made.

We have added a new paragraph in the Methods section where we compare our results with Baatsen et al. We find broad agreement in the climatology between our model and their model, with some differences in the behaviour of the overturning states. Note that Baatsen et al also find that the North Atlantic is too fresh to sustain sinking with an open Arctic-Atlantic gateway, and they find both on and off states of North Pacific sinking depending on the boundary / initial conditions used.

After the EOT, the Arctic gateway clearly opened again at some point, but the Arctic did not freshen and the AMOC didn't shut down again. Why was that? Was it because the Bering Strait opened, and that allowed the Arctic to become more saline?

This is an interesting question. At the EOT, the Arctic was extremely fresh so that its opening to the Atlantic inhibited overturning. This is obviously not the case in the modern ocean. The reasons for the fresh late-Eocene Arctic is discussed in Hutchinson et al. (2018) and relates both to the hydrological cycle and size of the Arctic. These things would have gradually changed as the Arctic expanded and the climate cooled. It would make for an interesting follow up study to examine the progressive salinification of the Arctic and find out at which salinity an opening to the Atlantic would allow for an AMOC. As to the Bering Strait, that is proposed to have opened much later than the Fram Strait opened so that is unlikely to be the cause for Arctic salinification at the EOT.

Can you hypothesise why the GSR mechanism appears to be important in the Miocene but not here? (or are the Starz et al simulations flawed in some way?)

We hypothesise that with Miocene-age paleogeography, the increased width of the Atlantic basin would strengthen the gyre circulation (and hence the northward salt transport) in the North Atlantic, and the opening of Fram Strait may increase the controlling effect of the GSR. In their model the GSR appears to be the shallowest 'barrier' between the basins, whereas in our simulations, the Barents Sea region is the shallow barrier. We have added some discussion on these possible effects.

We have also improved our illustration of the model bathymetry in Extended Data Figure 3, so that the changes to the Arctic gateways and the GSR are more clearly visible.

Specific comments:

Page 2, Line 5: Page 2, line 22: Figure 3 of Ferreira et al implies that the proxies indicate a more complex history of NADW than just "turning on" at the EOT. So would be good to moderate the language here somewhat.

We have changed this to "the onset or strengthening of an AMOC".

Page 3, Line 11: not sure why this is a "increase in temperature asymmetry". There is an asymmetry, yes, but not sure why it is increasing.

We have changed this to "this asymmetry in temperature evolution".

Page 4, line 1: "Geological reconstructions of the Barents Sea indicate that this region was the only viable connection between the Arctic and Atlantic in the late Eocene". What about east of Greenland? What about via Turgai Strait and paratethys?

This statement is now linked to Figure 1 which presents the Baatsen et al (2016) dataset and geological evidence for the Barents Sea uplift. On the Turgai Strait and Tethys connection, our experiments find that the freshwater outflow through this strait has no appreciable impact on Atlantic water masses. The narrow passageway between Greenland and Svalbard was a gridding artefact that was not present in the Baatsen et al (2016) dataset. Sensitivity tests indicate that closing this passageway with an open Barents Sea has minimal impact on salinity, and indeed our North Atlantic salinity is very similar to Baatsen et al (2018). These points are now explained on lines 109-117.

Page 6, line 22 “northward Ekman transport returns geostrophically” I am not sure what “returns” means in this context.

This has been changed to “returns southward in upper ocean western boundary currents”.

P7, line 16. Just because southern ocean gateway are uncertain, does not mean that they are less plausible as a mechanism.

This has been reworded. We emphasise that changes to the Southern Ocean gateways, and their resulting circulation, likely occurred over tens of millions of years, while the Arctic closure can occur rapidly due to the gateways being shallow and narrow.

Reviewer #3 (Remarks to the Author):

This manuscript reports a paleosimulation follow up study to Coxall et al., 2018 (NGS). The authors' aim is to investigate the cause of the inception of Atlantic overturning circulation (AMOC) at the Eocene Oligocene transition. They conclude in favour of Arctic closure as the trigger as opposed to the two more established hypotheses- deepening of the Greenland Scotland Ridge and development of wind-driven upwelling in the Southern Ocean.

We don't know that AMOC kicked in at the EOT but there are strong hints in published data sets that AMOC at least strengthened at that time. That's potentially a big deal because we do know that Antarctica grew its first large ice sheets at the EOT.

The manuscript falls short of establishing a mechanistic link between AMOC and Antarctic glaciation but that's okay. There's enough here to merit publication subject to some improvements outlined below. I think that these issues can be addressed without a big inflation in manuscript size because there are space savings to be made. For example the introduction could be tightened up by ditching the review text which goes back over elderly composite data sets/reviews. They could instead drive more rapidly to the key proxy evidence from the Atlantic for EOT AMOC start up.

We have restructured the introductory paragraphs so that the evidence of Atlantic sinking is more clearly grouped together. AMOC evidence, both from the EOT and the middle Eocene, is discussed in a single paragraph, while Pacific sinking is presented in a separate paragraph. We have trimmed a little overlap, but retain the contextual introduction of the EOT.

We need an improved project justification/articulation of the study's major advance. What, more precisely, is the original/significant result here? It is a useful step and important contribution to have shown that Arctic isolation promotes strong densification of and deep convection in the North Atlantic but, as the authors declare, that is not an entirely new result (ref 37 38). Given that the mechanism responsible for this change is gateway closure to the Arctic, it is not obvious why we can't confidently apply the findings from those Early Eocene

studies to the EOT. What are the differences, say in continental configuration/hydrological cycle between the Early Eocene and EOT or numerical models used that justify this new study and make its results unanticipated? A related issue is that we need a more prominent critical evaluation of the tectonic case for Arctic isolation around EOT time. There's no reference supporting the Barents Sea argument in the main text and too much of that discussion is hidden away in the Methods. What are the realistic uncertainties (including chronological) on the interpretations of refs 50-52?

The geological basis for conducting these experiments is now included in the main manuscript, and the accompanying figure is now Figure 1. We note that previous work on the role of Arctic closure was based on early Eocene paleogeography, which is substantially different from the late Eocene. The North Atlantic basin widens substantially from the early to late Eocene, and the possible sinking locations also change (as noted in response to Reviewer 1 on Nordic vs Labrador Sea sinking). It was therefore not obvious that the Arctic closure would lead to an AMOC startup using late Eocene paleogeography, nor that the AMOC would be shut down when the Arctic gateways are open. Furthermore, by testing our results in combination with the Southern Ocean opening and Greenland-Scotland Ridge hypotheses, we can argue that the Arctic Ocean gateway closure is more important than other factors that have been hypothesised to trigger the AMOC at the EOT.

The authors should tighten up on their terminology and address interesting results in their experiments (SAT) that contradict the present day. What exactly is a modern-like AMOC? It's a term used liberally in the manuscript but I can't find the authors' definition of what they mean. The text that comes closest is the discussion in paragraph-2 (bottom half of p.2) but it needs work. There's a garbled statement about composite stable isotope data sets and, in my view, it is ill-advised to state as a fact that modern-like AMOC developed at the EOT based on those data. More importantly, I find the result presented in Figure 3a interesting but doesn't it fundamentally contradict the idea that the inception/invigoration of AMOC at the EOT is in anyway modern-like? Today the northern hemisphere is warmer than the southern hemisphere largely because of AMOC-driven northward cross-equatorial heat transport. Figure 3a suggests that, if the continents are configured for the EOT, AMOC-driven warming of the circum-North Atlantic is completely counteracted by cooling in the North Pacific because of shutdown of its deep convection cell.

As noted above, we have clarified our description of the modern-like Atlantic structure to: "several Atlantic-wide compilations of stable isotopes suggest that a four-layer structure of water masses in the Atlantic, similar to the modern day, developed in the early Oligocene (Cramer et al. 2009; Katz et al. 2011; Borrelli et al. 2014). These analyses suggest that in the late Eocene, northern component water was confined to shallower depths and northern latitudes, whereas around the EOT, northern component water deepened and expanded across the equator."

The presence of an AMOC implies a significant heat and salt transport into the high latitudes of the North Atlantic. We attach plots below showing the northward heat transport for the atmosphere, global ocean, and its split into Atlantic and Indo-Pacific basins. These plots illustrate a substantial shift in northward heat transport from the Pacific to the Atlantic, but the global pattern remains largely unchanged. Northward atmospheric heat transport is also relatively unaffected. Our northward global heat transport is similar to modern-day estimates, so we do not see a contradiction to a modern-like AMOC. There are differences that arise in the tropics due to the open Panama gateway, so that the inter-basin comparison of heat transport becomes less 'clean' than in the modern case.

Note that we do not find an ocean state where northern hemisphere sinking is absent. To achieve a bipolar seesaw effect, there must be a switch between absence / presence of northern sinking (e.g. Broecker, 1998; Elsworth et al, 2017). Our results suggest that there was instead North Pacific sinking prior to the AMOC, hence no bipolar seesaw effect.

Figure 3b (P-E) is also interesting but under-explored in discussion. The authors focus a little obsessively on their P-E field results for the continents, with the aim I think to address the specific chain of events hypothesized by Ref 42 to link AMOC and Antarctic glaciation. But there appears to be a very nice result in their P-E simulations over the ocean that goes undiscussed. I see a strong result in the (sub)tropical belt, especially over the Pacific that appears diagnostic of an intensified/shifted ITCZ that merits more attention (cf. Hyeong et al., 2016).

We have now further investigated our results of the ITCZ position. In short, we find minimal changes in the ITCZ position, if anything a slight southward shift due to cooling of the North Pacific (since the ITCZ moves towards the warmer hemisphere). These changes are shown in Extended Data Figure 5. We note however, that Hyeong et al (2016) infer a northward shift of the ITCZ based on CO₂ cooling and glaciation of Antarctica at the EOT, two factors that we specifically do not include in these experiments. Overall, the shift from Pacific to Atlantic sinking would likely have a lesser effect on the ITCZ than the cooling of Antarctica.

Finally, I'd really like to see a further panel with polar projections for Figure 3 to better evaluate the footprint of changes in SAT and P-E on the real estate that became/did not become glaciated at the EOT (Antarctica/Arctic).

We have included polar projections of SAT and P-E over Antarctica, and included a comparison to halving CO₂, shown in Extended Data Figure 4. These show that the Antarctic cooling induced by the overturning change is small compared to the impacts of greenhouse forcing and glaciation.

Details

Sort out the line numbering, 1-25 on every page is painful.
The line numbering now continues across pages.

Bold para line 4, it is ambiguous
This sentence has been rewritten.

Bold para the last sentence needs to be bigger picture- what are the implications?
We have added a sentence on the climate implications of the circulation changes.

Page 3, para-1 this transition is ambiguous
We have reworded this to "a shift in sinking from the Pacific to the Atlantic".

Page 4, line 25 and overpage- sentence is garbled
This sentence has been clarified.

Paul A. Wilson

Reviewers' comments:

Reviewer #1 (Remarks to the Author):

Hutchinson et al., have thoroughly addressed all raised points with one exception:

The change of the Artic Gateway across the Barents shelf is plausibly presented as a mechanism for the initiation of a modern day AMOC. However, the geological evidence for the exact timing of this change at the EOT is still not entirely convincingly presented or slightly oversold. This could be tackled in two aspects:

(1) Figure 1c) I could not find this map in the three mentioned references. If it is taken from one of these, it should be made clearer, from which one. If it was produced for this manuscript, it should be made clearer, what the exact process was. Furthermore, Figure 1 is used to argue that this connection likely closed around the EOT. However, with only a snapshot of the E-O boundary, this cannot be assessed. How about a map of the early-middle Eocene? Screening through the references given for this Figure I could not find any indication that the seaway was open during the early Cenozoic. As the same mechanism has been proven to be also viable here, this is crucial.

(2) On some occasions the -AMOC onset vs. strengthening- still reads slightly one-sided. This could be avoided by calling it onset of a "modern-like AMOC" as done in other manuscripts (e.g. line 12).

Subject to the revisions of this aspect, I support publication in Nature Communications.

Maximilian Vahlenkamp

Reviewer #2 (Remarks to the Author):

See attached pdf.

Reviewer #3 (Remarks to the Author):

The authors have done a nice job overall revising their Ms. Most of the issues I raised are satisfactorily addressed but, while they responded to my question about why their result is unexpected in their letter, the Ms still lacks that clear message which is the more important thing. Furthermore, there appears now to be a related internal contradiction: line 124-129 vs. line 144 that should be clarified. Along the way, I've also noted two or three more minor wrinkles to straighten out.

Abstract wording is weird- sentence starting on line 13 starts we argue, the next sentence starts we test. In other words, as written, the interpretation comes before its supporting evidence. You might want to start the second of these two sentences: Our evidence comes from.

Line 53, ref 20 is out of place here, that study presented no sea surface temperature data

Line 54, 'is consistent with' would be better than 'supports'

Line 124-129, It is still not well explained in the Ms (although articulated in the rebuttal) why, when models show North Atlantic densification in response to Arctic closure for Early Eocene geography, the same result is unexpected for the Late Eocene. Furthermore, if the logic posited is that the result isn't obvious for a North Atlantic that is wider at the EOT than during the earlier Eocene then that logic seems contradicted by the hypothesis advanced in the sentence on line 144.

Line 144 (also response to R1 in rebuttal). Explain in a further clause/sentence why widening of the Atlantic might enable a larger region of deep water formation when less restriction might be expected to weaken salinity-driven forcing of densification (contradiction here to line 124-129).

Paul

We thank the reviewers for their positive feedback and further constructive comments on the manuscript. As before, we outline our response to each of the comments in blue text below.

Kind Regards,
David Hutchinson

Reviewer #1 (Remarks to the Author):

Hutchinson et al., have thoroughly addressed all raised points with one exception: The change of the Arctic Gateway across the Barents shelf is plausibly presented as a mechanism for the initiation of a modern day AMOC. However, the geological evidence for the exact timing of this change at the EOT is still not entirely convincingly presented or slightly oversold. This could be tackled in two aspects:

(1) Figure 1c) I could not find this map in the three mentioned references. If it is taken from one of these, it should be made clearer, from which one. If it was produced for this manuscript, it should be made clearer, what the exact process was. Furthermore, Figure 1 is used to argue that this connection likely closed around the EOT. However, with only a snapshot of the E-O boundary, this cannot be assessed. How about a map of the early-middle Eocene? Screening through the references given for this Figure I could not find any indication that the seaway was open during the early Cenozoic. As the same mechanism has been proven to be also viable here, this is crucial.

Figure 1c was redrawn from Smelror et al (2009), page 123. This has been made more explicit in the caption. We have also added a new Extended Data Figure 1, showing the Eocene reconstruction of Smelror et al (2009), and the early and late Eocene time slices from Lasabuda et al (2018, Mar. Pet. Geol., Fig. 15). We have also added the following sentence on Line 77-79, "Assemblages of silicoflagellates and ebridians from the middle Eocene show some similarities between North Atlantic and Arctic species, indicating an Arctic-Atlantic connection during the middle Eocene (Onodera et al, 2008)."

This complements the other independent evidence for at least early middle Eocene Nordic Seas-Arctic connections (already included in lines 71-77) based on the finding of Azolla freshwater algae extending from the Lomonosov Ridge to the North Seas between 49-48 Ma, and implying "... Arctic (-sourced) freshwater plumes to as far south as the southern North Sea basin" (Brinkhuis et al., 2006).

(2) On some occasions the -AMOC onset vs. strengthening- still reads slightly one-sided. This could be avoided by calling it onset of a "modern-like AMOC" as done in other manuscripts (e.g. line 12).

Line 12 has been adjusted to "onset or strengthening of the Atlantic meridional overturning circulation". As discussed in the previous revision, we adjusted the text to "onset or strengthening" on Line 31 and Line 54 when discussing previous proxy evidence, and we have now made that adjustment on Line 153 too. Furthermore, evidence of Atlantic sinking from the middle Eocene is discussed prominently from Lines 39-42, providing a balanced picture of this issue. The term "modern-like AMOC" is also imperfect and requires a detailed explanation, as was pointed out in the previous round of review. Thus we prefer to stick with "onset or strengthening" to cover both possibilities.

Subject to the revisions of this aspect, I support publication in Nature Communications.

Maximilian Vahlenkamp

Reviewer #2 (Dan Lunt) Remarks to the Author:

Many thanks to the authors for their comprehensive response to the reviewer comments. I read their rebuttal and was satisfied with their responses. I also re-read the manuscript, and have the following additional minor comments:

Line 77-80: A possible additional process may be related to changing geomorphology and tectonics leading to increased river routing towards the Arctic?

We have changed this to “enhanced hydrological cycle in the Eocene climate combined with topographically-induced river runoff”.

Line 69: In my opinion, this section (“Closure of the Arctic-Atlantic Gateway”) should only be concerned with the observational evidence for the gateway configuration. At present, it has a few sentences about the model setup. I think these model setup sentences should be moved to the beginning of the next section (“Ocean and climate response to Arctic closure”, or into methods). At the moment it is a bit odd that the model paleogeography is discussed before the model or simulations have even been introduced. I guess this is a hangover from when this section was in Supp Info?

We agree that these sentences about the model setup (previously Lines 111-117) are better placed in the Methods. We have combined them with our description of closing the Arctic gateway in the model (now Lines 295-301) in the Methods.

Line 131: This section should start with a call-out to the Methods section where the simulations and model are presented.

We have added an introductory sentence to this section with a reference to the methods.

Line 135: Before this you need to state which simulation you consider the “control” – open or closed gateway?

We have edited this to “a pre-EOT control simulation with an open Arctic-Atlantic gateway”.

Line 233: “must”? Another possibility is that although temperature warmed, precipitation increased substantially, building up snow and then an ice sheet.

This has been rephrased to “it must have occurred through indirect effects, such as circulation changes lowering atmospheric CO₂, or altering precipitation over Antarctica”.

Line 240: But not the only potential mechanisms, e.g. it may have been a decrease in CO₂ emissions.

This has been edited to “Atmospheric CO₂ can decrease due to many factors, such as a reduction in CO₂ emissions, or changes to geological sinks or carbon reservoirs. We consider here two potential mechanisms of atmospheric CO₂ reduction,…”

Line 262 – and including a weathering module.

We have rephrased this sentence to include “carbon cycle and weathering processes”.

Line 307-308: hasn't this just been said in the previous sentence? If not, please clarify.

This has been revised to: “Their ocean-only model simulations exhibit either North Pacific sinking, or South Pacific sinking, depending on the boundary and initial conditions⁴⁶, while their coupled model exhibits South Pacific sinking under the same paleogeography⁴⁷”.

I think that there could be a bit more discussion of Roberts et al (2009). In the intro you mention that they carry out similar simulations but for the early Eocene – towards the end of the manuscript it might be good to revisit this and suggest that, given that your results in essence agree with their simulations, the implication is that the role of the Arctic gateway opening/closure may be independent of other paleogeographic changes.

We have added a new paragraph to the “Existing hypotheses for AMOC start-up” discussion linking back to Roberts et al (2009) and Vahlenkamp et al (2018):

“Previous modelling studies have found that Arctic closure can trigger an AMOC under early Eocene paleogeography^{48,49}, however this is substantially different from the late Eocene. The North Atlantic basin widens over the 20 million years from the early to late Eocene, which affects the pathways of heat and salt transport to the possible sinking locations. In particular, a wider Atlantic basin creates a larger basin of wind forcing for the North Atlantic gyres and more prominent western boundary currents. It was therefore not obvious that the Arctic closure would trigger an AMOC using late Eocene paleogeography, nor that the AMOC would be shut down when the Arctic gateways are open. Furthermore, by testing our results in combination with the Southern Ocean opening and GSR hypotheses, we have demonstrated that the Arctic Ocean gateway closure is more important than other hypothesised triggers of the AMOC at the EOT.”

I think that it would be good to mention that, given the role of the salt-advection feedback, there may be some (as yet undiscovered) bistability in the model.

We have acknowledged this possibility at the end of the Methods section, “Given that our model exhibits distinct states of North Pacific and North Atlantic sinking, governed by salt-advection feedbacks, we cannot exclude the possibility of bistability. However, this would require significant further testing and re-equilibration of the model and is beyond the scope of this study.”

Reviewer #3 (Remarks to the Author):

The authors have done a nice job overall revising their Ms. Most of the issues I raised are satisfactorily addressed but, while they responded to my question about why their result is unexpected in their letter, the Ms still lacks that clear message which is the more important thing. Furthermore, there appears now to be a related internal contradiction: line 124-129 vs. line 144 that should be clarified. Along the way, I’ve also noted two or three more minor wrinkles to straighten out.

We have clarified the issues highlighted by the reviewer, and have incorporated the reasoning outlined in our rebuttal into the manuscript. See below.

Abstract wording is weird- sentence starting on line 13 starts we argue, the next sentence starts we test. In other words, as written, the interpretation comes before its supporting evidence. You might want to start the second of these two sentences: Our evidence comes from.

We have removed the words “we argue” and revised lines 13-16 to: “Proxy evidence of gradual salinification of the North Atlantic prior to the EOT and tectonically driven isolation of the Arctic suggest that closing the Arctic-Atlantic gateway could have triggered the AMOC at the EOT. We demonstrate this trigger of the AMOC using a new paleoclimate model...”

Line 53, ref 20 is out of place here, that study presented no sea surface temperature data

Ref 20 has been removed here.

Line 54, 'is consistent with' would be better than 'supports'

This change has been made.

Line 124-129, It is still not well explained in the Ms (although articulated in the rebuttal) why, when models show North Atlantic densification in response to Arctic closure for Early Eocene geography, the same result is unexpected for the Late Eocene. Furthermore, if the logic posited is that the result isn't obvious for a North Atlantic that is wider at the EOT than during the earlier Eocene then that logic seems contradicted by the hypothesis advanced in the sentence on line 144.

This issue is now resolved in the section "Existing hypotheses for AMOC start-up", using the argument we presented in the previous rebuttal:

"Previous modelling studies have found that Arctic closure can trigger an AMOC under early Eocene paleogeography^{48,49}, however this is substantially different from the late Eocene. The North Atlantic basin widens over the 20 million years from the early to late Eocene, which affects the pathways of heat and salt transport to the possible sinking locations. In particular, a wider Atlantic basin creates a larger basin of wind forcing for the North Atlantic gyres and more prominent western boundary currents. It was therefore not obvious that the Arctic closure would trigger an AMOC using late Eocene paleogeography, nor that the AMOC would be shut down when the Arctic gateways are open. Furthermore, by testing our results in combination with the Southern Ocean opening and GSR hypotheses, we have demonstrated that the Arctic Ocean gateway closure is more important than other hypothesised triggers of the AMOC at the EOT."

Line 144 (also response to R1 in rebuttal). Explain in a further clause/sentence why widening of the Atlantic might enable a larger region of deep water formation when less restriction might be expected to weaken salinity-driven forcing of densification (contradiction here to line 124-129).

We have added the following sentence:

"A wider Atlantic basin increases the area of wind forcing on the North Atlantic gyres, which may alter the heat and salt transport of the western boundary currents towards the deep water formation regions."

Our argument from Lines 124-129 (now Lines 120-125) is that this hypothesis needs to be tested under late Eocene boundary conditions, since that has never previously been done, there is substantial tectonic evolution between the early and late Eocene, and there are several competing hypotheses for what triggered the AMOC. We do not see a contradiction between these two sections.

Paul

REVIEWERS' COMMENTS:

Reviewer #1 (Remarks to the Author):

In my opinion the authors have addressed all outstanding questions satisfactory and I have no objections to publication at this point.

Kind Regards,
Maximilian

Reviewer #2 (Remarks to the Author):

I read the response to the reviewers and am satisfied with their responses.

Reviewer Comments

Reviewer #1 (Remarks to the Author):

In my opinion the authors have addressed all outstanding questions satisfactory and I have no objections to publication at this point.

We thank the reviewer for his positive recommendation.

Kind Regards,
Maximilian

Reviewer #2 (Remarks to the Author):

I read the response to the reviewers and am satisfied with their responses.

We thank the reviewer for his positive recommendation.